# The structure of an archaeal oligosaccharyltransferase provides insight into the strict exclusion of proline from the N-glycosylation sequon

Yuya Taguchi[1,3,4], Takahiro Yamasaki[1,4], Marie Ishikawa[1], Yuki Kawasaki[1], Ryuji Yukimura[1], Maki Mitani[1], Kunio Hirata [2] & Daisuke Kohda [1✉]

Oligosaccharyltransferase (OST) catalyzes oligosaccharide transfer to the Asn residue in the N-glycosylation sequon, Asn-X-Ser/Thr, where Pro is strictly excluded at position X. Considering the unique structural properties of proline, this exclusion may not be surprising, but the structural basis for the rejection of Pro residues should be explained explicitly. Here we determined the crystal structure of an archaeal OST in a complex with a sequon-containing peptide and dolichol-phosphate to a 2.7 Å resolution. The sequon part in the peptide forms two inter-chain hydrogen bonds with a conserved amino acid motif, TIXE. We confirmed the essential role of the TIXE motif and the adjacent regions by extensive alanine-scanning of the external loop 5. A Ramachandran plot revealed that the ring structure of the Pro side chain is incompatible with the $\phi$ backbone dihedral angle around $-150°$ in the rigid sequon-TIXE structure. The present structure clearly provides the structural basis for the exclusion of Pro residues from the N-glycosylation sequon.

[1] Division of Structural Biology, Medical Institute of Bioregulation, Kyushu University, Fukuoka, Japan. [2] RIKEN SPring-8 Center, Sayo, Hyogo, Japan. [3] Present address: MOLCURE Inc., AIRBIC A07, Kawasaki, Kanagawa, Japan. [4] These authors contributed equally: Yuya Taguchi, Takahiro Yamasaki.
✉email: kohda@bioreg.kyushu-u.ac.jp

Asparagine-linked (N-linked) glycosylation is one of the most ubiquitous post-translational modifications of proteins conserved in all domains of life. All eukaryotic and archaeal organisms have N-glycosylation systems[1–3]. N-glycans are oligosaccharides attached to Asn residues, and they affect various physicochemical properties of glycoproteins[4,5]. In their biological roles, the N-glycans function as tags recognized by other proteins and are involved in protein quality control and sorting inside cells, as well as cell–cell and host–pathogen interactions on cell surfaces[5,6]. N-Glycosylation is essential in Eukarya and the modern phyla of Archaea[7]. In contrast, in the ancient archaeal phylum Euryarchaeota, the N-glycosylation is dispensable for growth in laboratory conditions but presumably requisite for survival in harsh environments[8]. N-Glycosylation also occurs in the eubacterial genera Campylobacter[3] and Helicobacter[9], in which the N-glycosylation is non-essential for growth but important for virulence by promoting the adhesion of these human enteropathogenic bacteria to host cells[10]. Interestingly, some deep-sea-dwelling eubacteria also have the N-glycosylation system[11]. This fact implies that the horizontal gene transfer occurred multiple times independently from common ancestors of Eukarya and Archaea. Consequently, the eubacterial N-glycosylation systems have hybrid features of their eukaryotic and archaeal counterparts[2].

The oligosaccharide transfer occurs on the side-chain carboxamide group of the Asn residue in the N-glycosylation sequon, Asn-$X$-Ser/Thr, in polypeptide chains, where $X$ denotes any amino acid residue except for proline[12]. Hereafter, the residue position is defined as $X^{-2}$-$X^{-1}$-$Asn^0$-$X^{+1}$-Ser/Thr$^{+2}$-$X^{+3}$-$X^{+4}$-$X^{+5}$. Eubacteria use an extended 5-residue sequon[13], Asp/Glu$^{-2}$-$X$-Asn-$X$-Ser/Thr, although the presence of an acidic residue at position -2 is not absolutely required[14,15]. The amino acid bias in the middle position (position $+1$) of the N-glycosylation sequon is an interesting phenomenon. Statistical analyses of many glycosylated sites in glycoproteins revealed little preference for a particular amino acid at position $X$, except for the strict Pro exclusion in eukaryotic[16,17], archaeal[18], and eubacterial glycoproteins[19]. The N-oligosaccharyl transfer is catalyzed by an integral membrane enzyme, oligosaccharyltransferase (OST)[20,21]. The OST enzyme determines the non-preference and the exclusion of amino acid residues at position $+1$ of the glycosylated sequons[22,23]. To clarify the structural basis of the sequon selection rules, we need the three-dimensional structures of the OST enzymes in complexes with the two substrates, an oligosaccharide donor and an oligosaccharide acceptor. In contrast to the relatively invariable properties of the amino acid sequences of the acceptor sequon, the oligosaccharide donor is highly diverse among the three domains of life. The oligosaccharide donor has the general structure of lipid-phosphate(s)-oligosaccharide, and is thus referred to as a lipid-linked oligosaccharide (LLO). The lipid part is dolichol in Eukarya and Archaea, and polyprenol in Eubacteria[24,25]. Polyprenol is a long chain isoprenoid alcohol with the general formula, [α-terminus] HO-(CH$_2$-CH=C(CH$_3$)-CH$_2$)$_n$-H [ω-terminus], and dolichol is a special type of polyprenol that contains a saturated isoprene unit at the α-terminus. A diphosphate-type LLO is commonly used as the oligosaccharide donor for the OST catalyzed transfer reactions in the three domains of life, but a subset of Archaea, Euryarchaeota, exceptionally uses a monophosphate-type LLO[26–28]. The chemical structure of the oligosaccharide part is also diverse. Most eukaryotes use a well-conserved canonical 14-residue oligosaccharide structure, Glc$_3$Man$_9$GlcNAc$_2$, and lower eukaryotes use a shorter version of the 14-residue structure, lacking the terminal glucose and/or mannose residues[29]. In contrast, Archaea and Eubacteria use completely different sets of oligosaccharide structures from species to species, with respect to the number, composition, and branching pattern of the monosaccharides[8]. Considering the substantial divergence of the oligosaccharide donor structures, comparisons between distantly related OST enzymes can capture the essence of substrate recognition and enzyme catalysis.

The OST enzymes are hetero-oligomeric protein complexes in most eukaryotes, and single-subunit proteins in lower eukaryotes[20,21]. The archaeal and eubacterial OSTs are also single-subunit enzymes. The OST enzymes are located in the endoplasmic reticulum membranes of eukaryotic cells and the plasma membranes of archaeal and eubacterial cells. The crystal structures of the eubacterium Campylobacter lari OST (alias ClPglB) were reported in complexes with an acceptor peptide (PDB: 3RCE)[30], an acceptor peptide plus a non-hydrolyzable LLO analog (PDB: 5OGL)[31], and an inhibitory peptide plus a reactive LLO analog (PDB: 6GXC)[32]. The crystal structures of the euryarchaeon Archaeoglobus fulgidus OST (alias AfAglB) were determined in complexes with a sulfate ion, which mimics the phosphate group of LLO (PDB: 3WAJ)[33], and with an acceptor peptide (PDB: 5GMY)[23]. These binary and ternary complex structures provided many valuable insights into the oligosaccharyl transfer reaction. Recently, the cryo-EM single-particle structures of yeast OST and two human OST paralogs were reported[34–36]. The catalytic subunits, Stt3, in the multi-subunit OST enzymes have essentially identical structures to those of ClPglB and AfAglB[21]. The two human OST structures contain an endogenous dolichol-phosphate, which was co-purified during purification, and one of them also contains a model for an acceptor peptide of unknown origin. Unfortunately, the resolutions (3.3–3.5 Å) of the cryo-EM structures are not sufficient to discuss the details of the sequon recognition and the catalytic mechanism.

Several conserved short amino acid motifs have been identified in the diverse Stt3, AglB, and PglB protein sequences (identity <20%). The C-terminal globular domain contains the WWDYG and DK/MI motifs[37], where the slash delimiter indicates domain-specific conservation. The DK motif is found in Eukarya and a subset of Archaea, whereas the MI motif is present in the remaining Archaea and Eubacteria. The WWDYG and DK/MI motifs form a binding site for the Ser/Thr residue in the sequon[30,33]. The DGGK motif is conserved among eubacterial PglBs and euryarchaeal AglBs and presumed to be involved in LLO binding[38]. The equivalent in the eukaryotic Stt3 and the AglBs from the ASGARD and TACK superphyla of Archaea is a double sequon motif, DN$X$TZN$X$[T/S], where $X$ and $Z$ can be any residue[20]. The N-glycan attached to the double sequon motif is involved in the interactions with other subunits in the multi-subunit OST complexes[34–36]. The N-terminal transmembrane (TM) region of the Stt3/AglB/PglB proteins consists of 13 TM helices and contains two D$X$D motifs on the first and second external loops (EL1 and EL2) and a TIXE/SVSE motif on the fifth external loop (EL5). The TIXE motif is found in Archaea and Eubacteria, whereas the SVSE motif is present in Eukarya. The mutually independent conformational changes of the N-terminal and C-terminal halves of the EL5 loop are considered to be essential for the binding of the LLO and sequon, respectively[39,40]. In response to the conformational changes of the EL5 loop, the catalytic structure dynamically forms by integrating the Glu residue in the TIXE/SVSE motif. No appropriate functional groups were found around the side-chain carboxamide group of the acceptor Asn, and thus a hypothetical twisted amide mechanism was proposed for the activation of the inert amide nitrogen[30]. In this mechanism, the N–C bond in the carboxamide group is transiently twisted through bipartite interactions with the two carboxy groups of the conserved acidic residues in the first D$X$D and the TIXE/SVSE motifs. The twisting abolishes the conjugation of the lone-pair electrons on the nitrogen atom with

the carbonyl group, and thus increases the nucleophilic reactivity of the amide nitrogen[41].

Here, we determined the crystal structure of the ternary complex of the *A. fulgidus* AglB protein (*Af*AglB) with a sequon peptide and a dolichol-phosphate molecule. The catalytic structure around the bound metal ion is almost the same as that of the binary *Af*AglB-peptide complexes determined previously[23]. Our analysis of the sequon recognition revealed the special roles of the TIXE motif in the EL5 loop. Although the conservation of the TIXE/SVSE motif was previously reported, its precise role in the oligosaccharyl transfer reaction has not been identified. We now report the formation of the inter-chain hydrogen bonds between the sequon and the TIXE motif in the *Af*AglB protein. The requirement of a special $\phi$ dihedral angle in the rigid sequon-TIXE structure clearly explains the structural basis for the strict exclusion of Pro at the middle position of the N-glycosylation sequon.

## Results

**Crystallization and structure determination.** We used the lipidic cubic phase (LCP) method to obtain crystals of the *Af*AglB in a complex with a donor LLO molecule and an acceptor peptide. Native LLO was isolated from cultured *A. fulgidus* cells. The *Af*LLO preparation that produced diffraction-quality co-crystals was a crude mixture of LLOs with variable numbers of monosaccharides (6 and 7), isoprene units (C55 and C60), saturated isoprene units (3, 4, and 5), and the sulfate group (0 and 1)[27]. The peptide used for crystallization was custom synthesized. To compensate for the weak affinity, the sequon peptide was tethered to the *Af*AglB protein *via* a disulfide bond, to shift the association-dissociation equilibrium to the bound state[23]. A cysteine residue was introduced as a sole tethering point (G617C) in the *Af*AglB protein. To stop the transfer reaction, the Asn residue in the sequon was replaced by a L-2,4-diaminobutyrate (Dab) residue. The replacement of the amide group by an amino group is known to inhibit the oligosaccharyl transfer reaction in a competitive manner[42]. The peptide sequence is TAMRA-APY (Dab)VTAS*C*R-OH, in which the non-reactive sequon is underlined and the cysteine residue for tethering is italicized. The N-terminal $\alpha$-amino group is modified with a fluorescent carboxytetramethylrhodamine (TAMRA) dye for color detection. We chose 7.7 MAG as the host lipid with consideration of the larger water channel and reduced interfacial curvature of the cubic mesophase, which is suitable for the crystallization of membrane proteins with a large soluble domain[43]. Microcrystals were grown in a lipidic sponge mesophase under buffer conditions of 19–22% PEG400, 0.1 M Na-citrate, pH 6.0, and 50 mM NaCl. The positions and shapes of the crystals were easily identified by the magenta color of the TAMRA dye (Supplementary Fig. 1). Diffraction data were collected from 2529 microcrystals at the microfocus beamline BL32XU, SPring-8, Japan. A small-wedge data set was collected from each crystal and merged to complete the data set. The structure was determined by the molecular replacement method to a resolution of 2.7 Å (Table 1).

Even though the tethered peptide contained the non-reactive Dab residue at position 0, the LLO binding site was occupied by a dolichol-phosphate, instead of an intact LLO, suggesting that the LLO was hydrolyzed during the prolonged crystallization period. An omit electron density map revealed the clear densities for the dolichol-phosphate, except for the isoprene units in the middle part (Fig. 1d, orange mesh). Another omit electron density map also revealed the clear density for the sequon part in the tethered peptide (Fig. 1c, red mesh). Consequently, the models were reliably built for the sequon segment, A$^{-3}$PYDabVT$^{+2}$, and the dolichol(C60)-phosphate. The construction of the models of

the A$^{+3}$SC$^{+5}$ linker segment of the tethered peptide and the central part of the dolichol chain was guided by the chemical structures of the amino acids and isoprene unit. Three water molecules around the metal ion were visible in the difference map and modeled (Fig. 1c, blue mesh).

**Overall structure.** *Af*AglB comprises the N-terminal TM region (residues 6–498) and the C-terminal globular domain (residues 499–868). The flexible EL5 loop (residues 331–374) in the TM region is fully visible and constitutes integral parts of the binding sites for the substrate peptide and LLO (Fig. 1b). The comparison with the previous binary *Af*AglB-peptide complex[23] provides a dynamic view of the EL5 loop. The N-terminal half (residues 335-350) of the EL5 loop is disordered when the LLO binding site is vacant and becomes ordered to form an $\alpha$-helix in the presence of

**Table 1 Data collection and refinement statistics.**

| | Ternary complex of *Af*AglB, tethered peptide, and dolichol-P (PDB: 7E9S) |
|---|---|
| *Data collection statistics* | |
| Beamline | SPring-8 BL32XU |
| Wavelength (Å) | 1.0000 |
| Space group | $P2_12_12$ |
| Cell dimensions | |
| a, b, c (Å) | 345.74, 48.69, 63.56 |
| $\alpha$, $\beta$, $\gamma$ (°) | 90.0, 90.0, 90.0 |
| Resolution range (Å) | 49.40–2.70 (2.80–2.70) |
| No. of reflections | 30677 (3054) |
| $R_{merge}(I)$[a] | 0.432 (8.718) |
| $R_{meas}(I)$[b] | 0.435 (8.776) |
| $I/\sigma(I)$ | 13.6 (1.3) |
| $CC_{1/2}$ | 0.998 (0.756) |
| Completeness (%) | 100.0% (99.9%) |
| Redundancy[c] | 79.5 (76.9) |
| *Refinement statistics* | |
| Resolution range (Å) | 24.84-2.70 (2.796-2.70) |
| No. of reflections | 30599 (2954) |
| $R_{work}$[d] | 0.215 (0.300) |
| $R_{free}$[d] | 0.252 (0.357) |
| No. of non-hydrogen atoms | 7233 |
| Protein atoms | 7032 |
| Ligand atoms | 185 |
| Water molecules | 16 |
| rmsd[e] from ideal bond lengths (Å) | 0.003 |
| bond angles (°) | 0.60 |
| Ramachandran plot (%)[f] | |
| Favored region | 95.6 |
| Allowed region | 4.3 |
| Outlier region | 0.12 |
| Average B-factor (Å$^2$) | 79.0 |
| Protein atoms | 78.7 |
| Ligand atoms | 91.6 |
| Water molecules | 66.9 |
| No. of TLS groups | 3 |

Values in parentheses are for the highest resolution shell.
[a]$R_{merge}(I) = (\Sigma_{hkl}\Sigma|I_i - <I>|)/\Sigma_{hkl}\Sigma I_i$, where $I_i$ is the intensity of the *i*th observation and $<I>$ is the mean intensity. All sums over *hkl* extend only over unique reflections with more than one observation.
[b]$R_{meas}(I) = (\Sigma_{hkl}\sqrt{\frac{n}{n-1}}\Sigma|I_i - <I>|)/\Sigma_{hkl}\Sigma I_i$, where $I_i$ is the intensity of the *i*th of n observations, and $<I>$ is the mean intensity. $R_{meas}$ is a redundancy-independent version of $R_{merge}$.
[c]Although the high redundancy of reflections leads to the high $R_{merge}/R_{meas}$ values, the availability of more measurements leads to improved accuracy of the final data set.
[d]$R_{work}/R_{free} = \Sigma|Fo—Fc|/\Sigma|Fo|$. $R_{work}$ was calculated from the working set (95.1% of the total reflections). $R_{free}$ was calculated from the test set, using 4.9% of the total reflections. The test set was not used in the refinement.
[e]rmsd, root mean square deviation.
[f]Calculated using the program *MOLPROBITY*.

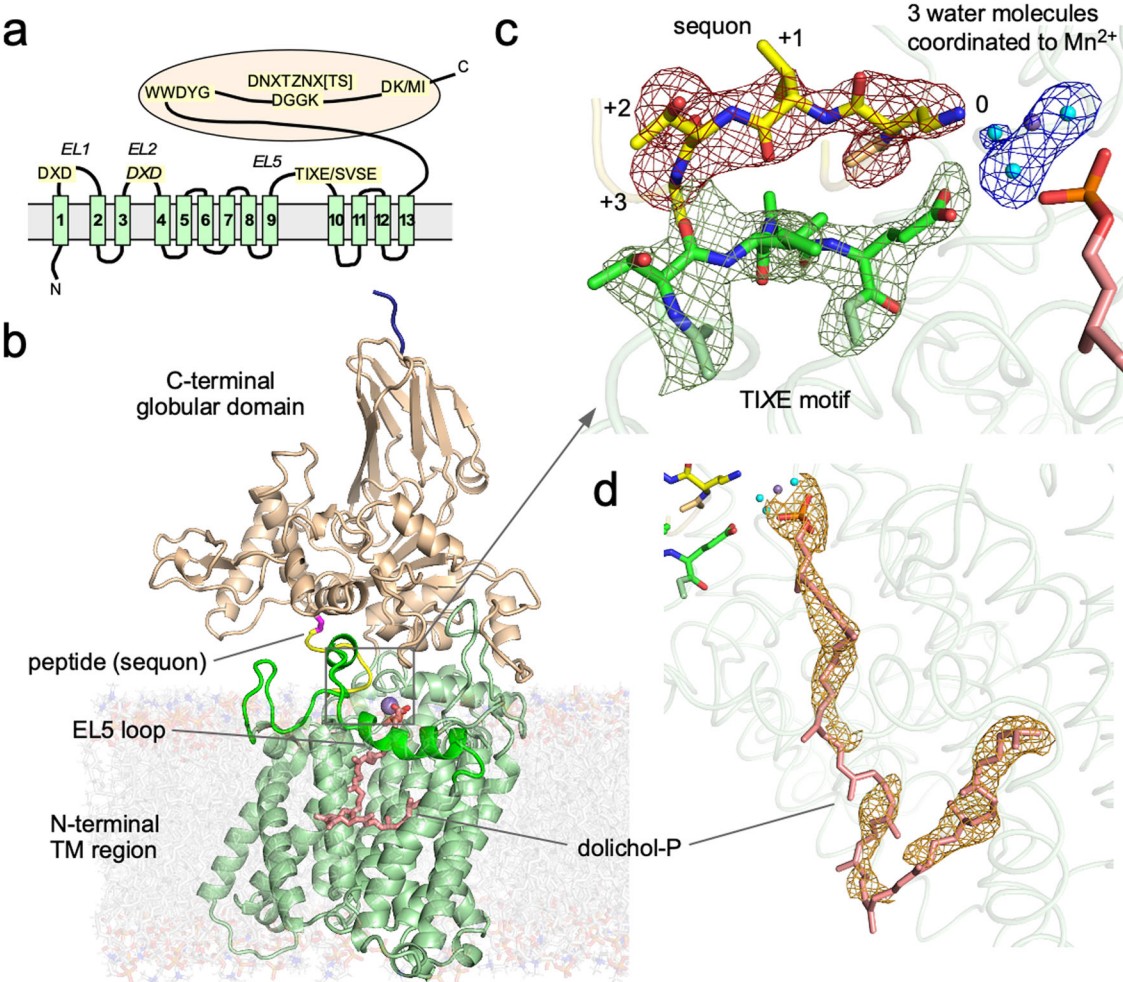

**Fig. 1 Structure of conserved amino acid motifs in the ternary complex. a** Positions of the short, conserved motifs on the amino acid sequence of the catalytic subunit, Stt3/AglB/PglB, of the OST enzyme. **b** Ribbon representation of *Af*AglB with the N-terminal transmembrane region in pale green and the C-terminal globular domain in wheat, embedded in a model DOPC lipid bilayer (transparent). The four extra residues in the spacer sequence, retained after the removal of the C-terminal His-tag, are modeled and colored dark blue. The EL5 loop is green, the $Mn^{2+}$ is shown as a purple sphere, and the dolichol-phosphate is shown as salmon sticks. The bound acceptor peptide is shown as a yellow loop and the disulfide bond for tethering is magenta. **c** Close-up view of the catalytic site. The red mesh depicts an omit electron density map of the sequon segment (yellow sticks). The numbers indicate sequon positions relative to the acceptor Asn. The green mesh depicts an omit map of the TIXE motif segment (green sticks). The dark blue mesh depicts an omit map of the three water molecules (cyan spheres) coordinated to $Mn^{2+}$. These three omit maps were calculated independently, after removing the relevant segment/molecules. Each map is only shown around the segment/molecules contoured at 3σ. The TAMRA dye at the N-terminus of the peptide was invisible in the maps, suggesting the absence of specific interactions with the protein. **d** Close-up view of the bound dolichol-phosphate. The orange mesh depicts an omit map contoured at 2σ and is only shown around the dolichol-phosphate.

dolichol-phosphate. Except for the conformational change of EL5, the overall structure of the *Af*AglB protein in the ternary complex is almost identical to that of the binary structure (rmsd 0.60 Å for 703 Cα atoms).

**Catalytic structure around the metal ion**. There are three water molecules around the metal ion, presumably $Mn^{2+}$, at distances of 1.9–2.5 Å (Fig. 2). The metal ion also interacts with the protein via coordination to the carboxylate group of Asp[47] in the first DXD motif (Gly-Asn-Asp), and the carboxylate group of Asp[161] and the imidazole group of His[163] in the second DXD motif (Asp-His-His) (Fig. 2b). These six ligands have a regular octahedral arrangement around the metal ion. The carboxylate group of Glu[360] in the TIXE motif (Thr-Ile-Ala-Glu) indirectly participates in the coordination to the metal ion through two of the three water molecules. The phosphate group of the

dolichol-phosphate also interacts with the metal ion indirectly, through another two of the three water molecules.

**Mutagenesis study of the EL5 loop**. We conducted alanine-scanning mutagenesis of the 44 continuous residues in the EL5 loop to identify the important residues for the oligosaccharyl transfer reaction (Fig. 3, Supplementary Figs. 2, 7, and 8, Supplementary Data 1). The three Ala residues in the 44 residues were replaced by Gly. In the N-terminal half of the EL5 loop, no critical residues were identified, whereas the C-terminal half showed sharp decreases of the activity in the segment Leu[356]-Phe[365]. This is a convincing result because this segment contains the conserved TIXE motif, Thr[357]-Ile-Ala-Glu[360].

**Attempts to uncouple the LLO hydrolysis activity from the oligosaccharyl transfer activity**. The present crystal structure

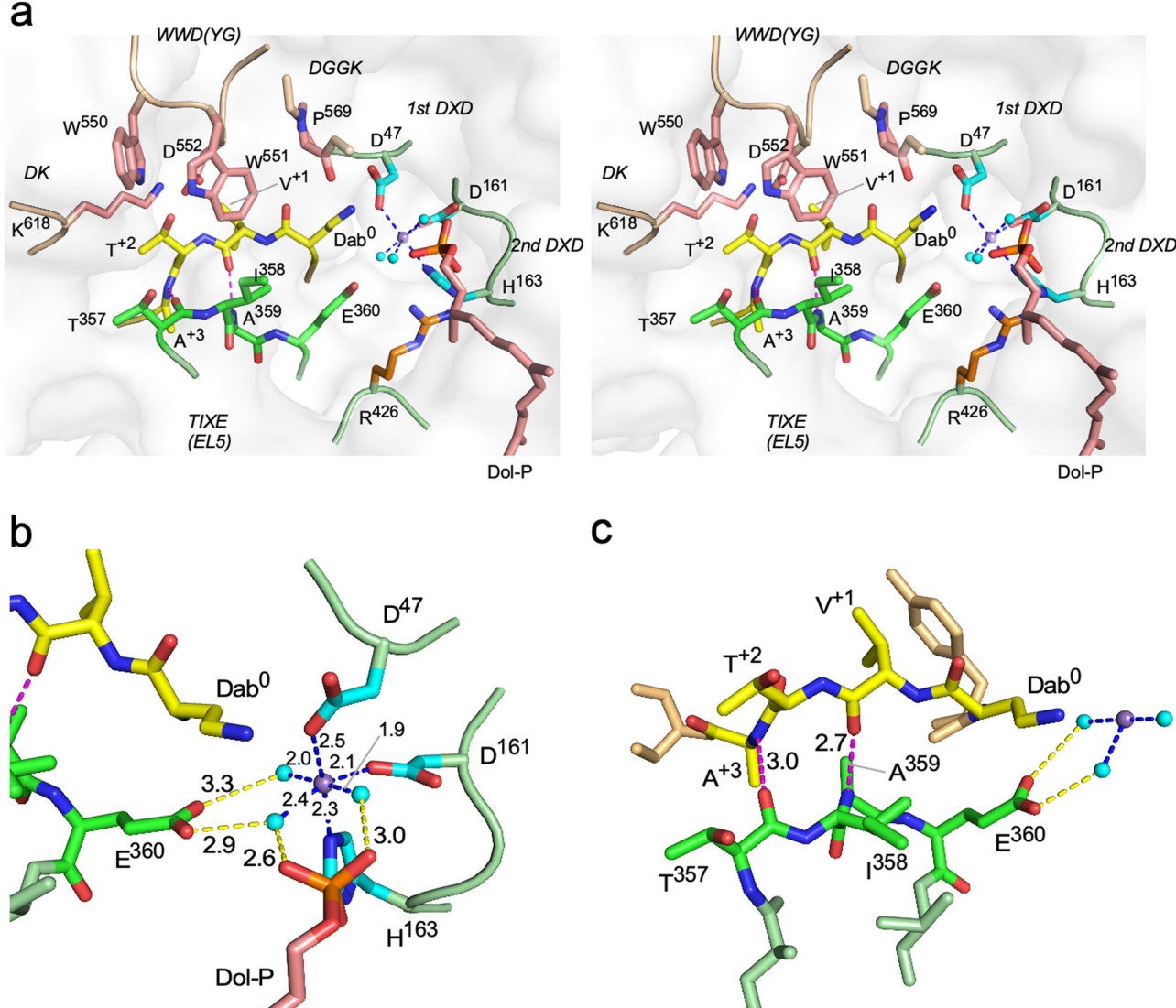

**Fig. 2 Close-up view of the catalytic site. a** Stereoview of the catalytic site. Stick representations are color-coded according to the conserved motifs: the two D*X*D motifs are cyan, the WWDYG, DGGK, and DK motifs are salmon, the TI*X*E motif is green, and the sequon is yellow. $Mn^{2+}$ is shown as a purple sphere, and the bound water molecules are cyan spheres. **b** Close-up view of the metal ion and the bound water molecules. **c** Anti-parallel β-sheet configuration between the sequon segment and the TI*X*E motif. In **b** and **c**, the dashed lines depict the interatomic interactions, and the numbers represent distances in Å units.

contains a dolichol-phosphate molecule instead of an intact LLO molecule. The eukaryotic and eubacterial OST enzymes are known to have hydrolytic activity that releases a free N-glycan (FNG) from LLO in the absence of a peptide substrate containing a sequon[44,45]. We have recently identified special point mutations of the yeast OST enzyme that uncouple the two activities[46]. Each mutation of Lys[586] and Met[590] in the DK motif (Asp-*XX*-Lys-*XXX*-Met) to Ala in the catalytic Stt3 protein subunit suppressed the LLO hydrolysis activity but retained the oligosaccharyl transfer activity. Interestingly, the DK motif is distant from the catalytic center (Fig. 2a). This indicates that the LLO hydrolytic activity can be remotely manipulated by modulating the peptide binding mode. Based on the yeast OST, we introduced point mutations in the *Af*AglB protein, expecting to obtain suitable mutants for crystallization with an intact LLO. The tested point mutations include His[81] and Arg[426] (close to the phosphate site), Asp[161] and His[162] (close to the metal site), and Asp[552], Gln[571], and Lys[618] (close to the peptide site). We measured the two enzymatic activities (Supplementary Figs. 3 and 9, Supplementary

Data 3). Disappointingly, no mutations with the desired suppression of the LLO hydrolysis activity relative to the oligosaccharyl transfer activity were obtained.

## Discussion

We determined the crystal structure of a ternary complex of the *Af*AglB protein with an acceptor peptide and dolichol-phosphate. The resolution of the present structure is one of the best (2.7 Å) among the available *Af*AglB and *Cl*PglB crystal structures (2.7–3.5 Å). The acceptor peptide was tethered to the *Af*AglB protein via an engineered disulfide bond[23]. The amino acid residues around the cysteine at the tethering point have high B-factors, indicating the minimal influence of the tethering on the sequon conformation. Although we used a non-reactive peptide containing an asparagine analog at position 0, a dolichol-phosphate molecule was bound in the complex, suggesting the hydrolysis of LLO. This is not surprising, considering that the eukaryotic and eubacterial OST enzymes have LLO hydrolytic

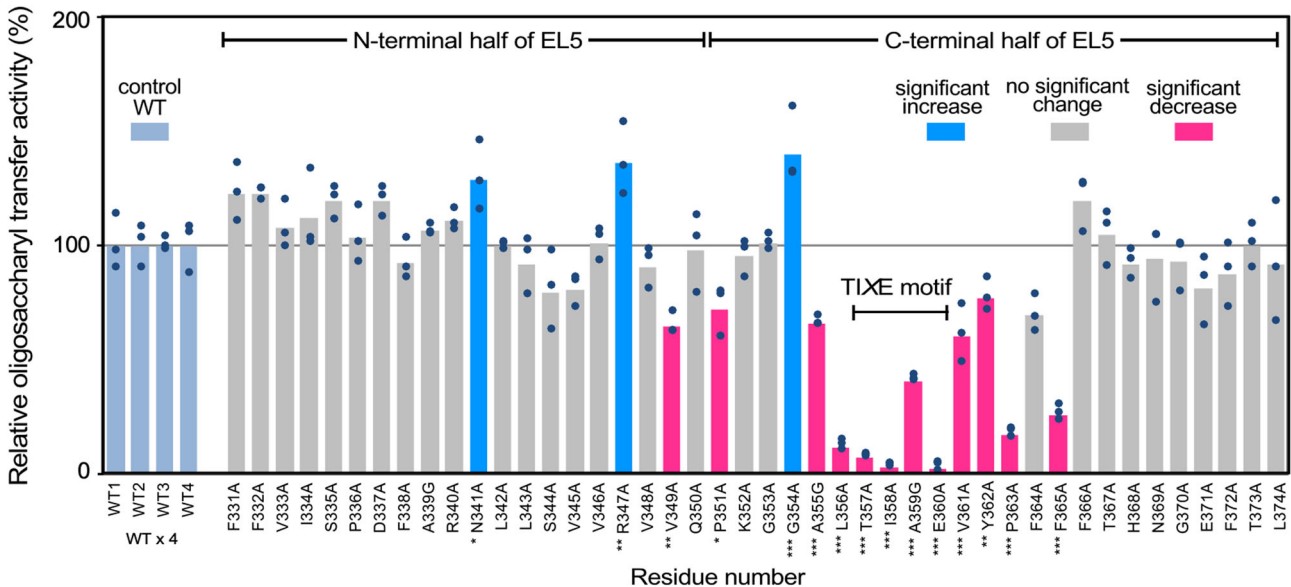

**Fig. 3 Alanine scanning mutagenesis of the EL5 loop.** The bar heights indicate the mean values of triplicate measurements in the oligosaccharyl transfer assay. Statistical analyses were conducted by one-way ANOVA followed by Dunnett's two-sided post-hoc test between each mutant and the corresponding wild type on the same PAGE gel. A total of 4 gels were used (Supplementary Figs. 2, 7, and 8). Adjusted $p$ values of 0.05 or less were considered statistically significant. *adjusted $p < 0.05$, ** <0.01, *** <0.001. The magenta bars represent the amino acid residues with a significant decrease, the blue bars represent those with a significant increase, and the gray bars represent those with no significant changes of the oligosaccharyl transfer activity upon mutation, whereas the pale blue bars represent the control reference of the wild-type enzyme.

activity to release free N-oligosaccharides[44,45]. Here, we showed that an archaeal OST/AglB also has the LLO hydrolytic activity (Supplementary Figs. 3 and 9, Supplementary Data 3).

The detailed structural comparison across the different domains of life yields a unified view of the structure and function of the OST enzyme. The functional structures are formed by conserved amino acid residues, mainly residing in short amino acid motifs such as the two D$X$D, TI$X$E/SV$S$E, WWDYG, DGGK/DN$X$TZN$X$[T/S], and DK/MI motifs (Fig. 1a). The spatial arrangements of the conserved residues are strikingly similar between the distantly related $Af$AglB and $Cl$PglB (Supplementary Fig. 4). The almost perfect superimposition suggests that the common substrate recognition and catalytic mechanisms were preserved through evolution.

The WWDYG and DK/MI motifs are involved in the formation of the Ser/Thr-binding pocket in the C-terminal globular domain, which explains the requirement of the hydroxy amino acid residues in the N-glycosylation sequon[30,33]. The sequon part, Dab-Val-Thr, plus the Ala residue at position +3 in the acceptor peptide, adopts an extended conformation and lies side-by-side in an antiparallel manner with the 4-residue TI$X$E motif, Thr-Ile-Ala-Glu, in the C-terminal half of the EL5 loop (Fig. 2c). Two inter-chain hydrogen bonds are formed between the carbonyl oxygen of Thr[357] and the amide group of Ala[+3], and between the carbonyl oxygen of Val[+1] and the amide group of Ala[359]. The hydrogen donor-acceptor distances are 2.7 Å and 3.0 Å, which are typical for moderate-strength hydrogen bonds. Although identical structures also exist in the previous $Af$AglB-peptide structure and $Cl$PglB-peptide-LLO structures (Supplementary Fig. 4), they were not mentioned explicitly at the time of publication[23,31,32], probably due to insufficient resolutions (PDB: 5GMY and 6GXC) or poor focus (PDB: 5OGL). The present structure has revealed that the $Af$AglB protein recognizes the sequon sequences through not only the side-chain groups of Asn[0] and Ser/Thr[+2], but also the main-chain groups of the $X$[+1] and $X$[+3] residues by the TI$X$E motif (Supplementary Fig. 4). The essential role of the TI$X$E motif was confirmed by the alanine-scanning study of the EL5 loop

(Fig. 3, Supplementary Data 1). No similar exhaustive mutation scanning experiments of the EL5 loop have been performed for other OST enzymes. Recognizing the structure and function of the TI$X$E motif is the key toward understanding the sequon recognition by the OST enzyme.

The exclusion of a Pro residue at position +1 in the N-glycosylation sequon is absolutely strict. No glycosylated Asn-Pro-Thr or Asn-Pro-Ser sites exist in the N-GlycositeAtlas database, an archive of more than 35,000 reviewed N-glycosylated sequences derived from human glycoproteins[47]. A significantly low level of glycosylation with Pro at position +3 has also been reported[16,17]. The unique structural features of Pro are easily presumed to be a possible cause of the exclusion. (1) The five-membered ring of Pro leads to a restricted $\phi$ dihedral angle around −75°. (2) Pro lacks a backbone amide hydrogen atom, which can be a donor for a hydrogen bond. (3) The $X$-Pro peptide bond tends to adopt a *cis* configuration. Consequently, proline destabilizes secondary structures and causes kinks in polypeptide chains.

The reason for the exclusion of Pro residues at positions +1 and +3 became obvious when the Ramachandran plots were generated (Fig. 4, Supplementary Data 2). All residues in the sequon sequence and the TI$X$E motif have $\phi$ values in the range from −60° to −120°, except for Val[+1] in the sequon, which has a high $\phi$ value around −150°. A similar high $\phi$ value of Ala[+1] in the eubacterial sequon sequence is also evident in the $Cl$PglB structure. The rigid ring structure of the side chain does not allow Pro to adopt such a high $\phi$ value and thereby excludes Pro from position +1 in the N-glycosylation sequon. The similar tendency of high $\phi$ values of Ala[+3] and PPN[+3] (para-nitrophenylalanine) in the two structures also accounts for the very low frequency of proline at position +3. Because the residue at position +3 is located at the boundary region of the rigid sequon-TI$X$E structure, the exclusion of Pro is probably less strict at position +3 than at the middle of the sequon.

Although $Af$AglB and $Cl$PglB catalyze the same reaction, $Af$AglB uses a mono-phosphate type LLO as the oligosaccharide

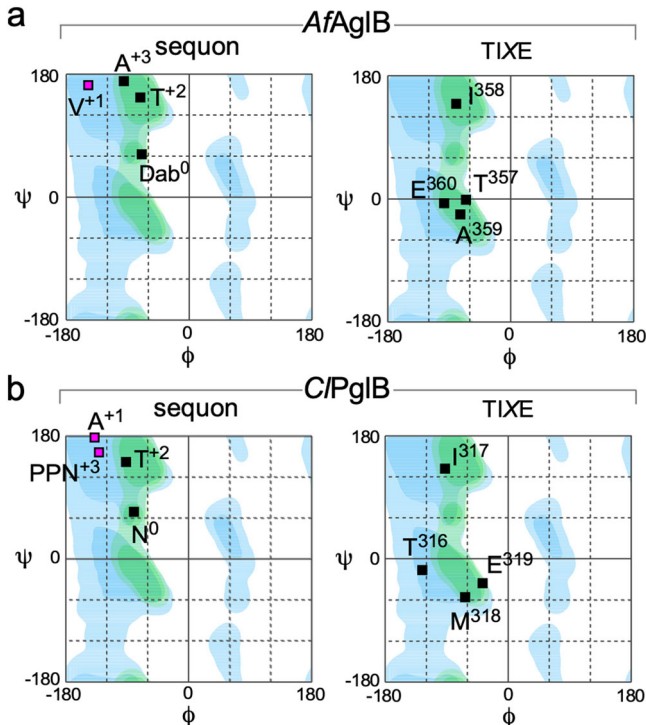

**Fig. 4 Ramachandran plots ($\phi$–$\psi$ diagrams) for the sequon segment and the TIXE motif. a** *Af*AglB and **b** *Cl*PglB (PDB: 5OGL). The favored and allowed regions for 19 amino acids are deep and pale blue, respectively, and the favored and allowed regions for proline are deep and pale green, respectively. PDB: 5OGL was used in consideration of its higher resolution (2.7 Å) than PDB: 6GXC (3.4 Å).

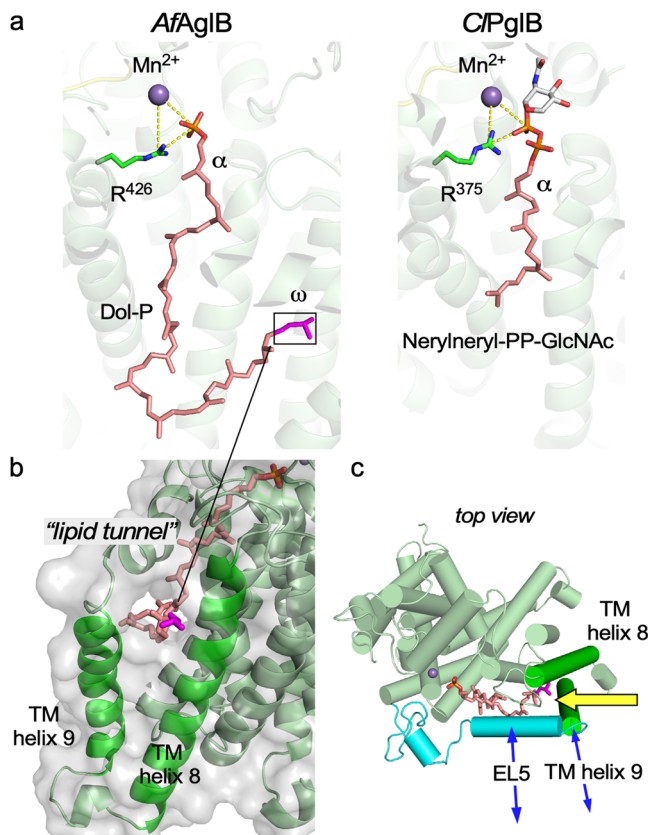

**Fig. 5 Interactions between *Af*AglB and bound dolichol-phosphate.**
**a** Dolichol-phosphate molecule bound to *Af*AglB and *Cl*LLO analog bound to *Cl*PglB (PDB: 6GXC). The tripartite relationships between the metal ions (purple spheres), conserved arginine residues (green sticks), and phosphate groups that directly bind to the reducing-end monosaccharide residues (orange and red sticks) are highlighted by the yellow dashed lines. **b** Lipid tunnel structure formed between TM helices 8 and 9. The ω-terminal isoprene unit is magenta. **c** Top view of the N-terminal TM region. The C-terminal globular domain is not displayed for clarity. TM helices 8 and 9 and two α-helices in the EL5 loop are depicted as green and cyan cylinders, respectively. The yellow arrow indicates the putative entry route of the *Af*LLO molecule into the binding site. The blue arrows indicate the motions between the closed (ordered) and open (disordered) states of TM helix 9 and the N-terminal half of the EL5 loop.

donor, whereas *Cl*PglB uses a di-phosphate type LLO. The lipid chains are also different. The lipid chain of *Af*LLO is dolichol, in which the α-terminal isoprene unit is saturated and contains a tetrahedral carbon with an *S*-configuration, while that of *Cl*LLO is polyprenol, which retains a double bond in the α-isoprene unit. Thus, comparative studies would provide insights into substrate recognition and enzymatic catalysis. We examined the correspondence relation of the phosphate groups in the *Af*AglB and *Cl*PglB structures. The phosphate group directly attached to the oligosaccharide moiety in *Cl*LLO is the counterpart of the single phosphate group in *Af*LLO (Fig. 5a). This implies a common catalytic mechanism for the two types of LLOs.

Next, we focus on the binding mode of the lipid chains. In the present *Af*AglB structure, the ω-terminus of the dolichol is in the tunnel structure formed at the interface between the two TM helices, TM helix 8 and helix 9 (Fig. 5b). This tunnel structure implies that the LLO molecule enters the binding site through the gap between TM helix 8 and helix 9 (Fig. 5c). TM helix 9 must move in concert with the conformational change of the EL5 loop, to enlarge the gap upon LLO binding. A similar mechanism, referred to as LLO entry gate, was proposed for the Stt3 subunit in the yeast OST, although no LLO molecule was bound in the determined cryo-EM structure[34]. The arrangements of the TM helix 8 and helix 9 are similar to each other between yeast Stt3 and *Af*AglB, but distinct in *Cl*PglB[21]. Consistently, for *Cl*PglB, the LLO was assumed to thread into the binding site under the disordered EL5, while the TM helix 9 stayed in place[31]. The mutagenesis of the Tyr[293] residue in the EL5 loop of *Cl*PglB resulted in a 7000-fold reduction of the glycosylation turnover rate[40]. By contrast, the V349A mutation, which is located at the corresponding position in *Af*AglB, exhibited a moderate reduction in the oligosaccharyl transfer activity, but the effect was not as

significant as in the *Cl*PglB case (Fig. 3). The discrepancy in the mutational effects is attributable to the different binding modes of the dolichol/polyprenol chains.

Finally, we discuss the yet-to-be-defined activation mechanism of the inert amide nitrogen in the acceptor Asn. To date, no convincing experimental evidence to support the twisted amide mechanism has been reported[32]. The OST enzymes might adopt a supportive mechanism to compensate for the poor nucleophilicity of the carboxamide group of the acceptor Asn. Locher and coworkers proposed that the divalent metal ion might directly activate the glycosidic oxygen to generate a reactive electrophile[32]. Alternatively, the rigid frame structure composed of the sequon and the TIXE/SVSE motif could function as a guiding device to bring the nitrogen atom in the vicinity of the C1 carbon atom (Supplementary Fig. 5). In the transition state, the amide nitrogen and the C1 carbon are forced to move within a closer reaction distance by the restriction of concerted motions to one direction. As the result, the unreactive amide nitrogen attacks the C1 carbon of LLO to

perform a nucleophilic substitution, by converting energy from the conformational to chemical coordinate[48].

In conclusion, the present structural and mutagenesis studies revealed the dual roles of the TIXE motif in the sequon recognition and catalytic mechanism. First, the TIXE motif participates in the formation of the rigid sequon-TIXE frame structure to recognize sequon sequences at the main-chain level (Fig. 2). The sequon-TIXE frame forces the amino acid residues at positions +1 and +3 to adopt high $\phi$ dihedral angles (Fig. 4), which are inaccessible to Pro. This is the structural basis for the exclusion of Pro residues at the middle position and the position after the Ser/Thr residue of the N-glycosylation sequon. As the second role, the rigid sequon-TIXE frame structure effectively restricts the motion of the acceptor Asn residue, which could compensate for the poor nucleophilicity of the carboxamide nitrogen (Supplementary Fig. 5).

## Methods

**Reagents.** *n*-Dodecyl β-D-maltopyranoside (DDM) was purchased from Dojindo. 1-(7Z-tetradecenoyl)-rac-glycerol (7.7 MAG) was purchased from Avanti Polar Lipids. The borane-dimethylamine complex and 2-aminopyridine (2-AP) were purchased from FujiFilm Wako Pure Chemical. The peptides were custom synthesized by Hayashi Kasei and Toray Research Center. A TAMRA fluorescent group was introduced during the peptide synthesis to either the N-terminus or the side chain of a lysine residue.

**Protein design for crystallization.** The binding affinity of a peptide containing a sequon is not sufficient for co-crystallization. To compensate for the weak affinity, we tethered a sequon peptide to the *Af*AglB protein via a disulfide bond. The cross-linked *Af*AglB-peptide was previously shown to be active in glycopeptide production in an intramolecular manner, which justifies the use of the tethered complex for structural studies. In the previous study, a cysteine residue at position +4 was used for tethering to a cysteine residue introduced by the G617C mutation in *Af*AglB[23]. For unknown reasons, the cross-linking level did not exceed 90%. In this study, we used position +5, since the final cross-linking level reached almost 100% (Supplementary Figs. 6 and 10, Supplementary Data 4). The Asn residue in the sequon was changed to an L-2,4-diaminobutyrate (Dab) residue. The replacement of the amide group by an amino group inhibits the oligosaccharyl transfer reaction in a competitive manner[42]. A Dab-containing peptide was successfully used in the crystal structure determination of *Cl*PgB with a reactive LLO analog[32]. The non-reactive peptide sequence we used is TAMRA-APY(Dab)VTASCR-OH, where the sequon is underlined and the cysteine residue for tethering is italicized. The N-terminal α-amino group is modified with a fluorescent carboxyte-tramethylrhodamine (TAMRA) dye for color detection, whereas the C-terminal carboxy group is unmodified. We confirmed the absence of oligosaccharyl transfer to the Dab-containing peptide, even in the state tethered to the *Af*AglB protein.

**Protein expression and purification.** The DNA encoding *A. fulgidus* AglB-L (AglB-L, the longest homolog of the three AglB proteins) was amplified from the genomic DNA, and subcloned into pET-52b(+) (Novagen) between the NcoI and SacI sites. The amino acid sequence (868 residues) is available through UniProtKB as UniProt entry O29867(AGLB3_ARCFU). An inverse PCR-based site-directed mutagenesis kit (SMK-101, TOYOBO) was used to generate single-point mutations of the *Af*AglB sequence. The transformed *Escherichia coli* C43 (DE3) cells (Lucigen) were grown at 310 K to an $OD_{600}$ of 0.8–1.0 in Terrific Broth, supplemented with 100 mg L$^{-1}$ ampicillin. Then, isopropyl β-D-1-thiogalactopyranoside was added at a final concentration of 0.5 mM. After 18-h induction at 291 K, the cells were harvested by centrifugation, and disrupted by sonication in 50 mM Tris·HCl, pH 8.0, 100 mM NaCl. The membrane fractions were collected by ultracentrifugation at $100,000 \times g$ for 2 h, and solubilized in the same buffer containing 1% (w/v) DDM (Dojindo). After ultracentrifugation at $100,000 \times g$ for 1 h, the recombinant protein in the supernatant was purified by affinity chromatography on nickel Sepharose High Performance resin (GE Healthcare) in the buffer containing 0.1% DDM. The *Af*AglB protein was expressed with a C-terminal His$_{10}$-tag after a thrombin cleavage site. For crystallization, the His-tag was cleaved by thrombin after affinity chromatography. Consequently, the protein contains an extra 7-residue spacer sequence, ELALVPR, at the C-terminus. After the protein was concentrated with an Amicon Ultra-4 device (100 kDa NMWL) (Merck Millipore), gel filtration chromatography using Superdex 200 10/300 GL (GE Healthcare) was performed in 20 mM Tris-HCl, pH 8.0, 300 mM NaCl, and 0.05% (w/v) DDM. For disulfide bond tethering, purified *Af*AglB(G617C) was incubated with a peptide at pH 8.0, at a molar ratio of 1:10. After an overnight incubation at room temperature, the *Af*AglB−peptide complex was separated from the unreacted peptide monomers and the byproduct peptide dimers by membrane filtration using an Amicon Ultra-4 (100 kDa NMWL) and concentrated to 33 mg mL$^{-1}$ by membrane filtration, in 20 mM Tris-HCl, pH 7.5, 200 mM NaCl, and 0.05% DDM.

For the oligosaccharyl transfer and FNG generation assays, the *Af*AglB mutant proteins were purified by nickel affinity chromatography only, and the His-tag at the C-terminus was not removed.

**Lipid-linked oligosaccharide from *A. fulgidus* cells.** *Archaeoglobus fulgidus* strain DSM 4304 (NBRC 100126) was obtained from the NITE Biological Resource Center (Tokyo). The cells were grown anaerobically without shaking, at 80 °C for 3 days. The culture medium was a simplified version of the predefined medium (NBRC Medium No. 1019)[49]. The cells were collected by centrifugation and disrupted in hypoosmotic buffer, consisting of 20 mM Tris-HCl, pH 7.5, and 2 mM MgCl$_2$. The cell disruption solution was supplemented with benzonase (Novagen) and complete protein inhibitor mixture, EDTA-free (Roche). The cell suspension was centrifuged at $8500 \times g$ for 15 min, and the supernatant was discarded. The pellet was suspended in the buffer and homogenized with a probe sonicator. After centrifugation for debris removal, the supernatant was ultracentrifuged at $100,000 \times g$ for 2 h to collect the membrane fractions. The membrane pellets were stored at −80 °C until use. The membrane pellets were resuspended in 20 mM Tris-HCl, pH 7.5, containing 0.1 M NaCl. Aliquots were transferred into glass round-bottomed centrifuge tubes. Methanol and chloroform were added to yield a methanol:chloroform:membrane fraction ratio of 2:1:0.8. The mixture was centrifuged to collect the clarified supernatants. Chloroform and water were added to the supernatants to yield a chloroform:water:supernan ratio of 1:1:3.8. The lower clear organic phase, containing the LLO, was collected and dried under a nitrogen gas stream in a draft chamber, redissolved in a small volume of CHCl$_3$:CH$_3$OH:H$_2$O 10:10:3 (v/v/v) (CMW), and stored in glass containers at −20 °C. The resultant crude *Af*LLO is a mixture of LLOs with variable numbers of monosaccharides (6 and 7), isoprene units (C55 and C60), saturated isoprene units (3, 4, and 5), and a sulfate group (0 and 1)[27].

The crude *Af*LLO was further separated by anion exchange chromatography with a HiTrap DEAE FF column (GE Healthcare), which was equilibrated with CMW containing 3 mM acetic acid. The absorbing materials were eluted with CMW containing 0.3 M ammonium acetate. The LLO concentrated by the two-phase partitioning was further separated by normal phase HPLC on a SUPELCO column (581513-u, Sigma-Aldrich), with a linear gradient from 100% solvent A CHCl$_3$:CH$_3$OH:NH$_4$OH 800:195:5 (v/v/v) to 100% solvent B CHCl$_3$:CH$_3$OH:H$_2$O:NH$_4$OH 450:450:95:5 (v/v/v/v). The coincidence between the peaks in the TIC (total ion chromatogram) and the oligosaccharyl transfer activity indicated the high purity of the LLOs in the fractions from the HPLC column. The collected *Af*LLO fractions were dried in a SpeedVac concentrator, redissolved in a small volume of CMW, and stored in glass containers at −20 °C. The purified *Af*LLO has a fixed number of monosaccharides (6 or 7) and a sulfate group (1), but has variable numbers of isoprene units (C55 and C60) and saturated isoprene units (3, 4, and 5)[27].

**LCP crystallization and microcrystal harvesting.** The purified cross-linked *Af*AglB-SS-peptide was reconstituted into a lipidic mesophase formed by the 7.7 MAG lipid. In the crystallization screening, we tested native LLO preparations of different purity grades isolated from cultured *A. fulgidus* cells. When we used purified LLO preparations, which were eluted as a single peak from a normal phase HPLC column, many microcrystals of fine appearance were obtained, but the quality of their X-ray diffractions was poor. We then switched to crude LLO preparations, prepared only by two-phase partitioning with a chloroform/methanol/water solvent system, and found that the microcrystals provided good diffraction data. Consequently, the *Af*LLO in the crystallization drops was a mixture of LLOs with variable numbers of monosaccharides (6 and 7), isoprene units (C55 and C60), saturated isoprene units (3, 4, and 5), and a sulfate group (0 and 1)[27]. The *Af*LLO dissolved in CMW was mixed with the melted 7.7 MAG lipid. The volume ratio was calculated to adjust the molar ratio of protein to *Af*LLO to 1:2–3 in the final cubic phase sample. The extra CMW was evaporated from the mixture fluid with a SpeedVac concentrator. The *Af*AglB-SS-peptide was mixed with the lipid mixture fluid, using a coupled syringe-mixing device at a protein solution/lipid mixture fluid ratio of 1:1 (w/w) at room temperature. Fifty-nanoliter (30-nL in a smaller scale) drops of the cubic phase sample were dispensed onto a lower film plate (Diffrax, Molecular Dimensions) and overlaid with 0.75 μL (0.45 μL) of precipitant solutions with a Crystal Gryphon LCP (Art Robbins Instruments). The plates were sealed with a thin upper film and stored at 20 °C. Initial crystallization screening was performed using MemGold, MemGold2, and MemStart+MemSys kits (Molecular Dimensions). Crystals were obtained under the following precipitation conditions: 0.1 M Li$_2$SO$_4$, 0.1 M glycine, pH 9.5, 24–30% PEG400; 50 mM NaCl, 0.1 M Na-citrate, pH 6.0, 18–22% PEG400; 0.2 M Na-citrate, 0.1 M Tris-HCl, pH 8.8, 26–30% PEG400. Crystals appeared after 1 or 2 days and grew to full size in a week. The microcrystals were needle-shaped with a length greater than 100 μm and a width/thickness less than 5 μm. Individual compartments that contained microcrystals grown in 50 mM NaCl, 0.1 M Na-citrate, pH 6.0, 18–22% PEG400 were excised with a homebuilt punching device, "AOMUSHI". The positions and shapes of the crystals were easily identified by the magenta color of the TAMRA dye attached to the peptide substrate (Supplementary Fig. 1). The 6-mm-diameter pieces of the bilaminar films were mounted on homemade acrylic cryoloops with grease as an adhesive, flash cooled in liquid nitrogen, placed in Unipucks (Crystal Positioning Systems, New York), and stored in liquid nitrogen.

**X-ray data collection and data processing**. The X-ray diffraction data were collected at beamlines BL32XU and BL44XU in SPring-8 (Hyogo, Japan). The final X-ray diffraction data were collected at beamline BL32XU using an EIGER X 9M detector (Dectris, Switzerland). A micro-focused beam of 10 μm × 15 μm (horizontal × vertical) with a wavelength of 1.0000 Å was used for both the raster scan and data collection, under a cryo stream operating at 100 K. The datasets in 10° wedges were collected from microcrystals with a frame rate of 50 Hz in a shutterless operation mode at a dose of 10 MGy. The automated data collection system ZOO, developed at SPring-8[50], was used for automatic data collection from 2529 microcrystals supported on 5 cryoloops. Data sets indexed with consistent unit cell parameters were subjected to a hierarchical cluster analysis based on unit-cell similarity. Finally, 483 datasets were integrated, merged, and scaled to 2.7 Å using the automatic data processing system KAMO[51]. KAMO is an open-source data-processing pipeline, which utilizes existing programs, including the XDS[52] and BLEND[53].

**Structure determination**. The program phenix.phaser in PHENIX was used for the initial phase determination[54], by molecular replacement with the structure of the revised version of the *Af*AglB-*SS*-Ac-RYNVTAC-NH$_2$ structure (PDB: 5GMY) as the search model, after removing the EL5 loop and the tethered peptide. The asymmetric unit contained one protein molecule. Crystallographic refinement was performed with the program phenix.refine in PHENIX[55] and the program REFMAC5 in CCP4[56]. Further manual model rebuilding was performed with the program COOT[57]. The TLS refinement was done with three TLS groups, consisting of the N-terminal TM region (residues 6-494 of chain A), the C-terminal globular domain (residues 495-872 of chain A), and the bound peptide (residues 1-9 of chain B). The N-terminal 5 residues, MQNAE, and C-terminal 3 residues, VPR, were not modeled. In the previous structure determinations of the OST proteins, a metal ion bound to the catalytic site of the AglB/PglB/Stt3 proteins was assumed to be Mg$^{2+}$, Mn$^{2+}$, or Zn$^{2+}$ contained in the crystallization solutions as an ingredient[23,30,32,33]. Mn$^{2+}$ and Mg$^{2+}$ have been suggested as the physiological cation of the OST enzymes[58]. In the present structure, Mn$^{2+}$ was selected as an endogenous metal ion, because no metal ions were explicitly added in the crystallization solution. *Fo*—*Fc* maps suggested that Mg$^{2+}$ had too few electrons, but Mn$^{2+}$ fitted well. The extra nonprotein densities were modeled as the dolichol-phosphate (1 molecule), 7.7 MAG (4 molecules), PEG (di-hydroxyethyl ether, 5 molecules), and bound water (16 molecules). The chemical structure of the dolichol moiety used for modeling had twelve isoprene units (C60), of which three units were saturated on the ω-terminal side. The program eLBOW in PHENIX was used to generate the CIF restraints of the dolichol-phosphate, 7.7 MAG, and PEG by using the SMILES strings, CC(C)CCCC(C)CCCC(C)CCCC(\C)=C/CCC(\C)=C/CCC(\C)=C/CCC(\C)=C/CCC(/C)=C\CC\C(C)=C\CC/C(C)=C\CC/C(C)=C\CC[C@H](C)CCO[P](O)(O)=O, CCCCCC\C=C/CCCCC(=O)OC[C@H](O)CO, and OCCOCCO, respectively. Data collection and refinement statistics are summarized in Table 1.

The *Af*AglB crystal structure was embedded in a model DOPC lipid bilayer generated by the Membrane/Bilayer builder in the CHARMM-GUI program[59]. Figures were generated with PyMOL, version 2.3 (Schrödinger). Ramachandran plots were generated with the program RAMPAGE (CCP4 supported program)[60].

**Revision of the previous *Af*AglB-peptide complex structure**. We previously reported a crystal structure of the *Af*AglB-peptide complex (PDB: 5GMY)[23]. The metal ion was assumed to be Mg$^{2+}$, which was derived from the crystallization solution. During the present study, we noticed that the metal ion was misplaced at one of the bound waters. This could be a consequence of the same number of electrons in H$_2$O and Mg$^{2+}$. The corrected coordinates were used in the molecular replacement in the present study as the search model, after removing the EL5 loop and the tethered peptide. Data collection and refinement statistics of the revised coordinates are provided in Supplementary Table 1. The revised version of the coordinates was deposited with the same PDB entry name, 5GMY, using the PDB entry versioning system.

**Oligosaccharyl transfer assay**. The oligosaccharyl transfer assay was performed by the PAGE method, as described previously[61,62]. The reaction mixture comprised the wild-type or mutated *Af*AglB protein (3 nM), an acceptor peptide substrate (3 μM), and crude *Af*LLO (0.2 μM) in a 10 μL reaction solution, containing 100 mM Tris-HCl, pH 7.5, 10 mM MnCl$_2$, and 0.02% (v/v) Tween 20. The acceptor peptide is Ac-AAYNVTKRK(TAMRA)-OH, in which a fluorescent TAMRA dye is attached to the side-chain amino group of the C-terminal Lys residue for detection. The requisite amount of *Af*LLO in a chloroform/methanol/water solvent was dried and re-dissolved in the reaction solution, which contained Tween 20 to solubilize the LLO. The reaction was performed in an oven at 65 °C for 30 min or 1 h. To stop the reaction, 5 × SDS sample buffer was added. The in-gel fluorescence images of the SDS–PAGE gels were recorded with an LAS-3000 multicolor image analyzer (Fuji Photo Film), with green LED excitation.

**FNG generation assay**. The wild-type or mutated *Af*AglB protein (0.1 μM) was incubated with crude *Af*LLO (5 μM) at 65°C for 1 h in a 100 μL reaction solution, containing 100 mM Tris-HCl, pH 7.5, 5 mM MnCl$_2$, and 0.02% (v/v) Tween 20.

The requisite amount of LLO in a chloroform/methanol/water solvent was dried and re-dissolved in the reaction solution, which contained Tween 20 to solubilize the LLO. The reaction was terminated by the addition of 10 μL of 0.2 M EDTA-NaOH, pH 8.0. A 330 μL aliquot of ethanol was added, and the reaction solution was incubated for 15 min at 4 °C. After centrifugation at 15,000 × *g* for 15 min, the supernatant was evaporated to dryness. The dried oligosaccharides were dissolved in 500 μL of water and loaded on a PD MiniTrap column G-25 (GE Healthcare) for desalting. The column was preequilibrated with 5% ethanol before use. The fraction containing oligosaccharides was collected and the eluted oligosaccharides were evaporated to dryness. The reducing end of the dried oligosaccharides was derivatized with 2-AP. The dried oligosaccharides were incubated with 20 μL of 2-AP in acetic acid at 80 °C for 1 h. After the reaction, the mixture was incubated with 20 μL of a borane-dimethylamine complex in acetic acid, at 80 °C for 30 min. The excess 2-AP was removed using a MonoFas silica gel spin column (GL Sciences). The spin column was washed with water and then preequilibrated twice with 800 μL of 100% acetonitrile before use. The sample solution was mixed with 460 μL of 100% acetonitrile and loaded onto the spin column. The column was washed twice with 800 μL of 95% (v/v) acetonitrile. Water was added to the column to elute the fluorescently labeled oligosaccharides. The pyridylaminated oligosaccharides were separated by HILIC (hydrophilic interaction liquid chromatography) chromatography with an AdvanceBio Glycan Mapping column (Agilent Technologies), using an Infinity 1290 UPLC system (Agilent Technologies) equipped with an in-line fluorescence detector. Solvent A was 100 mM ammonium acetate buffer, pH 4.5, and solvent B was 100% acetonitrile. The column was equilibrated with 20% solvent A at a flow rate of 0.5 ml min$^{-1}$. The gradient cycle was 12 min with a 1-min isocratic segment at 20% solvent A, a 1-min linear gradient to 27% A, a 6-min linear gradient to 40% A, a 1-min isocratic segment at 100% A, and a 3-min isocratic segment at 20% A.

**Statistics and reproducibility**. Statistical analyses were performed with the EZR statistical software[63], an open-source statistical software program based on R and R commander[64]. Data were analyzed by one-way analysis of variance (ANOVA) followed by Dunnett's two-sided post hoc test for multiple comparisons.

**Reporting summary**. Further information on research design is available in the Nature Research Reporting Summary linked to this article.

## Data availability
The atomic coordinates of *Af*AglB-*SS*-TAMRA-APY(Dab)VTASCR in a complex with dolichol-phosphate have been deposited in the PDB, with the accession code 7E9S. The source data underlying Figs. 3, 4, Supplementary Figs. 3a, b, and 6c are provided in Supplementary Data 1–4. The uncropped gel images related to Fig. 2a, b, Supplementary Figs. 3d, and 6d are provided as Supplementary Figures.

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

## Acknowledgements

We thank Drs. Tadashi Suzuki and Yoichiro Harada for advice on the FNG generation assay, and Mr. Seiichiro Hayashi and Ms. Hisano Yajima for mass spectrometry of the reaction mixture for the preparation of the *Af*AglB-peptide complex. The DNA sequencing service was provided by the Laboratory for Research Support at the Medical Institute of Bioregulation at Kyushu University. The experiments at beamline BL44XU, SPring-8, were performed under the Cooperative Research Program of the Institute for

Protein Research of Osaka University, as Proposals 20196914 and 20206514. The experiments at beamline BL32XU, SPring-8, were supported by the Platform Project for Supporting Drug Discovery and Life Science Research (Basis for Supporting Innovative Drug Discovery and Life Science Research, BINDS) from AMED under Grant Number JP19am0101070 to K.H., and by JSPS KAKENHI Grant Number JP21H02448 to D.K.

## Author contributions

Y.T., T.Y., M.I., Y.K., R.Y., and M.M. performed experiments under supervision by K.H. and D.K. Y.T. performed protein and LLO preparations, LCP crystallization, X-ray data collection, and structure refinement calculations. T.Y., M.I., Y.K., and R.Y. performed alanine-scanning mutagenesis, protein purification, and oligosaccharyl transfer assays. M.M. performed mutagenesis, oligosaccharyl transfer, and FNG generation assays. K.H. developed the ZOO system and contributed to the structure refinement calculation. Y.T. and D.K. wrote the paper.

## Competing interests

The authors declare no competing interests.
