## [Peer Review File · Communications Biology]

Reviewers' Comments:

Reviewer #1:

Remarks to the Author:

N-glycosylation is a major post-translational modification. It is very important to understand the detailed mechanism of the transfer reaction as catalyzed by OST. Numerous structures of OST have been published. But this manuscript is unique and valuable because the structure is determined to a good resolution of 2.7 Å, in which both donor and acceptor substrates are captured. The structure reveals that the acceptor peptide binding chamber is lined by the WWDYG and DGGK motifs on the ceiling and the TIXE motif on the floor. Importantly, the +1-position valine forms a backbone hydrogen bond with the crucial TIXE motif. Due to the backbone hydrogen bonding, the valine side chain points outwards, explaining the "X" (any residue) of the acceptor motif, but at the same time, also explains why proline cannot be at +1 site, because proline does not allow H-bond with TIXE. This insight is valuable, and I believe the MS is suitable for publication in Communications Biology. A few minor comments below.

1) The Rmerge(I) and Rmeans(I) are 0.432(8.718) and 0.435 (8.776) (Table 1). They are high. The author should briefly explain this, perhaps due to the use of exceedingly small crystals and the merge of diffraction data from a large number of crystals?

2) The density map of 3 water molecules coordinating the Mn²⁺ looks less definitive (Fig. 1, panel C). Mn²⁺ should be 2 times stronger than water. Please discuss or offer possible explanation. The author should also clearly note that the assignment of Mn²⁺ is based on literature as they did not provide direct evidence.

3) In page 13-14: "The present structure has revealed that the AfAgIB protein recognizes the sequon sequences through not only the side-chain groups of Asn0 and Ser/Thr+2". Please present a panel showing "AfAgIB protein recognizes the sequon sequences through the side-chain groups of Asn0 and Ser/Thr+2"?

Reviewer #2:

Remarks to the Author:

Manuscript ID# COMMSBIO-21-0878-T

Manuscript Title: Structural basis for the strict exclusion of proline from the N-glycosylation sequon

CONTRIBUTING AUTHORS: Yuya Taguchi et. al

Corresponding author: Daisuke Kohda

The authors have determined the crystal structure of a ternary complex of the oligosaccharyltransferase (OST) enzyme of the euryarchaeon, *Archaeoglobus fulgidus* (AfAgIB). In the ternary complex, the sequon peptide was tethered to the AfAgIB protein through a disulfide bond, and contained a dolichol phosphate. This group has previously determined the crystal structures of AfAgIB protein in complex with acceptor peptide or with sulfate ion (to mimic the donor LLO substrate).

The article is well-written and clear. This work is of importance in the understanding of N-linked glycosylation reaction. It is of interest to the scientific community. Although several high-resolution crystal structures AfAgIB and PglB (from eubacterium *Campylobacter lari*) in complex with the peptide sequon or inhibitor peptide have been solved previously (by this group and others), the analysis of the phi and psi angles for the peptide in the complex using Ramachandran's plot was never carried out.

In this article, the authors have analyzed the phi and psi angles for the peptide sequon in the complex with the Ramachandran's plots (Fig. 4) and have come up with the rationale for the strict exclusion of the proline residue in the sequon based on the torsion angle values. This rationale is sound and certainly explains the exclusion of proline in the peptide sequon.

The second important finding is that through alanine scanning, the C-terminal half of the EL5 loop has been shown to be critical for the enzyme activity.

However, there are few concerns that are listed below-

(1) To prepare the peptide tethered to AfAgIB, the authors incubated the protein with the peptide at pH 8.0 at 1:10 ratio. After overnight incubation, the AfAgIB-peptide complex was separated from the unreacted peptide monomers and byproduct peptide dimers.

- (a) The detail of the separation of the mixture of products of this reaction is not provided.
- (b) How did the authors know that there are peptide dimers?
- (c) Were the products of this reaction characterized by mass spectrometry?

Reviewer #1 (Remarks to the Author):

N-glycosylation is a major post-translational modification. It is very important to understand the detailed mechanism of the transfer reaction as catalyzed by OST. Numerous structures of OST have been published. But this manuscript is unique and valuable because the structure is determined to a good resolution of 2.7 Å, in which both donor and acceptor substrates are captured. The structure reveals that the acceptor peptide binding chamber is lined by the WWDYG and DGGK motifs on the ceiling and the TIXE motif on the floor. Importantly, the +1-position valine forms a backbone hydrogen bond with the crucial TIXE motif. Due to the backbone hydrogen bonding, the valine side chain points outwards, explaining the “X” (any residue) of the acceptor motif, but at the same time, also explains why proline cannot be at +1 site, because proline does not allow H-bond with TIXE. This insight is valuable, and I believe the MS is suitable for publication in *Communications Biology*. A few minor comments below.

1) The $R_{\text{merge}}(I)$ and $R_{\text{meas}}(I)$ are 0.432(8.718) and 0.435 (8.776) (Table 1). They are high. The author should briefly explain this, perhaps due to the use of exceedingly small crystals and the merge of diffraction data from a large number of crystals?

=> As you say, the high R_{merge} and R_{meas} values are a consequence of merging data from 2500 individual crystals into a single data set. These initial high- R_{merge} values become increasingly insignificant as each data set supports the determination of high-quality atomic coordinates. This fact is mentioned occasionally, such as 1) Soares AS et al. “Solvent minimization induces preferential orientation and crystal clustering in serial micro-crystallography on micro-meshes, in situ plates and on a movable crystal conveyor belt”. *J Synchrotron Radiat.* 21:1231-9, 2014. and 2) Wlodawer A et al., “Protein crystallography for aspiring crystallographers or how to avoid pitfalls and traps in macromolecular structure determination”. *FEBS J.* 280:5705-36, 2013.

In the revised manuscript, we added a note in the legend of Table 1 as follows: Although the high redundancy of reflections leads to the high $R_{\text{merge}}/R_{\text{mean}}$ values, the availability of more measurements leads to improved accuracy of the final data set.

2a) The density map of 3 water molecules coordinating the Mn^{2+} looks less definitive (Fig. 1, panel C). Mn^{2+} should be 2 times stronger than water. Please discuss or offer a possible explanation.

=> If the metal ion was deleted in addition to the three water molecules around the metal ion for the generation of a $F_o - F_c$ omit map in Fig.1, panel c, the electron density close to the metal ion became too strong and masked the three water molecules. Thus, we only deleted the coordinates of the three water molecules for clarity. Thus, no electron density corresponding to the metal ion exists. We guess that the reviewer misinterpreted the weak electron density around the metal ion due to the

non-optimal view angle. The view from a different angle might be better, but we would like not to change the present view angle because the other parts become difficult to see with different view angles.

2b) The author should also clearly note that the assignment of Mn^{2+} is based on literature as they did not provide direct evidence.

=> As you suggested, we added a sentence, " Mn^{2+} and Mg^{2+} have been suggested as the physiological cation of the OST enzymes" on page 20, with a citation, Gerber S et al., Mechanism of bacterial oligosaccharyltransferase: in vitro quantification of sequon binding and catalysis. J Biol Chem. 288:8849-61, 2013.

3) In pages 13-14: "The present structure has revealed that the AfAglB protein recognizes the sequon sequences through not only the side-chain groups of Asn⁰ and Ser/Thr⁺²". Please present a panel showing "AfAglB protein recognizes the sequon sequences through the side-chain groups of Asn⁰ and Ser/Thr⁺²"?

=> We added two dotted circles to emphasize the two interactions with the Asn⁰ and Ser/Thr⁺² in the supplementary Fig. 4. Accordingly, we added a sentence, "The previously identified interactions between the AfAglB/C/PglB proteins and the side-chain groups of Asn⁰/Dab⁰ and Thr⁺²/Thr⁺² are shown enclosed in dotted circles." in the legend of Supplementary Fig. 4 (page 33). In the main text, we cite Supplementary Fig. 4 in the appropriate position (page 13, line 2).

Reviewer #2 (Remarks to the Author):

The authors have determined the crystal structure of a ternary complex of the oligosaccharyltransferase (OST) enzyme of the euryarchaeon, *Archaeoglobus fulgidus* (AfAglB). In the ternary complex, the sequon peptide was tethered to the AfAglB protein through a disulfide bond and contained a dolichol phosphate. This group has previously determined the crystal structures of AfAglB protein in complex with acceptor peptide or with sulfate ion (to mimic the donor LLO substrate).

The article is well-written and clear. This work is of importance in the understanding of N-linked glycosylation reaction. It is of interest to the scientific community. Although several high-resolution crystal structures AfAglB and PglB (from eubacterium *Campylobacter lari*) in complex with the peptide sequon or inhibitor peptide have been solved previously (by this group and others), the analysis of the phi and psi angles for the peptide in the complex using Ramachandran's plot was

never carried out.

In this article, the authors have analyzed the phi and psi angles for the peptide sequon in the complex with Ramachandran's plots (Fig. 4) and have come up with the rationale for the strict exclusion of the proline residue in the sequon based on the torsion angle values. This rationale is sound and certainly explains the exclusion of proline in the peptide sequon.

The second important finding is that through alanine scanning, the C-terminal half of the EL5 loop has been shown to be critical for the enzyme activity.

However, there are few concerns that are listed below-

(1) To prepare the peptide tethered to AfAglB, the authors incubated the protein with the peptide at pH 8.0 at 1:10 ratio. After overnight incubation, the AfAglB-peptide complex was separated from the unreacted peptide monomers and byproduct peptide dimers.

(a) The detail of the separation of the mixture of products of this reaction is not provided.

=> We used membrane filtration for the separation of the AfAglB-SS-peptide/AfAglB from excess unreacted peptides.

We added a phrase, "by membrane filtration using an Amicon Ultra 100K device (Merck Millipore)" on page 18, lines 3-4.

(b) How did the authors know that there are peptide dimers? and (c) Were the products of this reaction characterized by mass spectrometry?

=> Since the peptide contains a cysteine residue for disulfide bond cross-linking, the formation of a cross-linked dimer of the peptide is unavoidable due to air oxidation. To address the comments, we re-purified the AgAglB(G617C) protein and performed the disulfide bond tethering experiment to prepare the AfAglB-SS-peptide (TAMRA-APY(Dab)VTASCR) complex. We performed MALDI-TOF-MS analysis to detect the dimer peak of the peptide. The expected m/z value was observed within 0.14 of the theoretical value. We uploaded the MS data for your reference as material for review only. We expected to observe a peak corresponding to the AfAglB-SS-peptide complex, but neither a peak corresponding to the AfAglB nor AfAglB-SS-peptide complex was observed. This is an understandable result because the ionization of such a large protein molecule is hard to occur in the presence of detergent (DDM). The MS/MS analysis of the glycosylated product in the cross-linked complex was reported in the previous study (Biochemistry 2017, 56, 602-611,

DOI: 10.1021/acs.biochem.6b01089), but it makes no sense in this study because the peptide used is unreactive. We established the preparative method of the disulfide bond tethered complex for crystallography in the previous study (*ibid.*). We believe that the full range characterization of the products is not necessarily required in this study. We hope that our reply will convince the reviewer.

Reviewers' Comments:

Reviewer #1:

Remarks to the Author:

The authors have satisfactorily addressed my concerns. I have no further critiques.

Reviewer #2:

Remarks to the Author:

The modification that the author have added now to the manuscript regarding the separation of protein:peptide complex is fine.